# Effect of Supplementation of a Cryopreservation Extender with Pectoliv30 on Post-Thawing Semen Quality Parameters in Rooster Species

**DOI:** 10.3390/antiox13081018

**Published:** 2024-08-21

**Authors:** Esther Díaz Ruiz, Juan Vicente Delgado Bermejo, José Manuel León Jurado, Francisco Javier Navas González, Ander Arando Arbulu, Juan Fernández-Bolaños Guzmán, Alejandra Bermúdez Oria, Antonio González Ariza

**Affiliations:** 1Department of Genetics, Faculty of Veterinary Sciences, University of Córdoba, 14071 Cordoba, Spain; v12dirue@uco.es (E.D.R.); id1debej@uco.es (J.V.D.B.); aarando@abelur.eus (A.A.A.); 2Institute of Agricultural Research and Training (IFAPA), Alameda del Obispo, 14005 Cordoba, Spain; 3Agropecuary Provincial Centre, Diputación Provincial de Córdoba, 14071 Cordoba, Spain; jmlj01@dipucordoba.es; 4Instituto de la Grasa, Consejo Superior de Investigaciones Científicas (CSIC), 41013 Sevilla, Spain; j.fb.g@csic.es (J.F.-B.G.); aleberori@ig.csic.es (A.B.O.)

**Keywords:** alperujo, antioxidant, avian local breed, pectin, rooster, sperm cryopreservation

## Abstract

Sperm cryopreservation is a fundamental tool for the conservation of avian genetic resources; however, avian spermatozoa are susceptible to this process. To cope with the high production of reactive oxygen species (ROS), the addition of exogenous antioxidants is beneficial. Pectoliv30 is a substance derived from alperujo, and in this study, its effect was analyzed on seminal quality after its addition to the cryopreservation extender of roosters at different concentrations. For this purpose, 16 Utrerana breed roosters were used, and seminal collection was performed in six replicates, creating a pool for each working day with ejaculates of quality. After cryopreservation, one sample per treatment and replicate was thawed, and several seminal quality parameters were evaluated. Statistical analysis revealed numerous correlations between these variables, both positive and negative according to the correlation matrix obtained. Furthermore, the chi-squared automatic interaction detection (CHAID) decision tree (DT) reported significant differences in the hypo-osmotic swelling test (HOST) variable between groups. Moreover, results for this parameter were more desirable at high concentrations of Pectoliv30. The application of this substance extracted from the by-product alperujo as an antioxidant allows the improvement of the post-thawing seminal quality in roosters and facilitates optimization of the cryopreservation process as a way to improve the conservation programs of different endangered poultry breeds.

## 1. Introduction

Semen cryopreservation is currently a method that allows the ex situ conservation of genetic resources. However, this process causes various cellular lesions, with avian spermatozoa being particularly sensitive due to their high level of polyunsaturated fatty acids (PUFAs), which sensitizes them to reactive oxygen species (ROS) and lipid peroxidation (LPO) [1,2]. In this regard, avian seminal plasma contains several antioxidant enzymes to cope with ROS, but during cryopreservation processes, there is a considerable increase in these enzymes, and an imbalance is generated, leading to oxidative stress and plasma membrane peroxidation, which impacts sperm functionality by compromising mitochondrial activity, ATP production, and, thus, sperm motility, which ultimately negatively influences fertility [3,4].

In order to cope with the increase in ROS, the hypothesis that the addition of different antioxidants to cryopreservation extenders, whether natural or synthetic, can counteract the negative effect of ROS has been studied in recent years [5].

Alperujo is a by-product resulting from the biphasic extraction of olive oil and is a combination of liquid and solid waste. It is one of the main by-products generated in the agri-food industry, with approximately 5 million tons produced every year in Spain. The production of these large quantities of alperujo has negative environmental consequences since this product has phytotoxic compounds and a high organic content. The use of this by-product in the procurement of high-value compounds can be of great interest for waste reduction [6]. This substance contains phenols, carbohydrates, and proteins of high added value. Moreover, pectin polysaccharides can be obtained from the aqueous fraction by its thermal treatment, which results in the separation of the solid and liquid phases and the solubilization of compounds in the liquid phase [7,8,9].

Pectins are complex heteropolysaccharides that have prebiotic properties and anti-inflammatory activity and regulate the intestinal passage. When pectins are treated with pH adjustment, they give rise to modified pectins with a molecular weight of about 10 kDa. These modified pectins called Pectoliv have good physico-chemical properties. Also, Pectoliv extracts have antioxidant properties that are directly related to their antiproliferative activity. In this way, Pectoliv extracts can reduce the concentration of ROS, due to their free-radical-scavenging capacity [10].

In addition, modified pectins have other beneficial health effects as they possess anti-inflammatory and prebiotic properties and play an important role in the control of diabetes and the prevention of cancer and obesity [6]. These beneficial effects are related to binding to galectin-3, a lectin involved in tumorigenesis and cancer progression [11].

An example of a modified pectin is Pectoliv30, whose chemical composition is shown in Table 1. Given its antioxidant properties and the fact that this molecule is a polysaccharide that can act as a non-permeable cryoprotectant, this study hypothesized that the addition of Pectoliv30 to a seminal extender can have beneficial effects on the quality of post-thawed semen in roosters.

The Utrerana avian breed originates from Andalusia and is a locally adapted rustic breed that is mainly exploited for egg laying in extensive systems [12]. However, due to the low census of existing specimens, which as of 31 December 2023 was around 1300 [13], this breed is currently classified as endangered according to the Spanish Royal Decree-Law 45/2019. To enhance the situation of this breed, several morphological and productive characterization studies have been carried out in recent years [12,14,15,16,17,18,19,20,21]. However, less information about the improvement of assisted reproductive techniques related to this breed can be found today [22,23,24,25,26].

Taking into account that within Spain, Andalusia produces 80% of the total olive oil [27] and the fact that its by-products can be used to optimize conservation programs for the indigenous breeds that coexist in the olive grove implies a lower impact on the environment and is also beneficial from the point of view of human health and animal welfare [28].

For the evaluation of semen quality in the avian species, different techniques include not only more routine tests, such as mass motility, morphology, and membrane functionality (hypo-osmotic swelling test, HOST) [26,29,30], but also other more specialized techniques, such as the computer-assisted sperm analyzer (CASA) system and flow cytometry [31]. By applying statistical tools, we can predict which semen quality parameters provide the most information when evaluating thawed rooster semen. Through data mining, relationships or patterns existing in large data sets can be determined, which allows decomposing such data, providing useful information [32]. Within data mining, the chi-squared automatic interaction detection (CHAID) decision tree (DT) method is widely used to perform classification, prediction, regression, estimation, description, visualization, and dimensionality reduction, belonging to the non-parametric category [33]. This tool was successfully applied in the field of poultry breeding in another study evaluating the effect of hydroxytyrosol (HT) on thawed rooster sperm after addition to a cryopreservation diluent, providing interesting information [23]. Likewise, through its application, researchers have been able to identify the most important predictors of difficult calving in cattle [34]. In addition, this statistical tool has been applied in other areas of knowledge, such as animal production, where it has allowed classifying eggs of different genotypes of local breeds of hens according to egg quality traits [16], as well as determining which environmental factors lead to higher milk production in cows [35].

Therefore, this study aimed to evaluate different in vitro semen quality parameters to study the effect of the addition of different concentrations of Pectoliv30 to a cryopreservation extender in frozen-thawed semen of the Utrerana rooster breed through the application of the CHAID decision tree method.

## 2. Materials and Methods

### 2.1. Ethics Statement

The Spanish Ministry of Economy and Competitiveness, through the Royal Decree-Law 53/2013 and its accredited entity, the Ethics Committee of Animal Experimentation from the University of Córdoba, granted permission for the application of the protocols used in this study. These protocols were cited in the fifth section of the second article of the decree, as the animals assessed were used for accredited zootechnical purposes. This national decree follows the European Union Directive 2010/63/UE dated 22 September 2010.

### 2.2. Animal Management

In this study, 16 Utrerana breeder roosters between 1 and 3 years of age were used. They were located in the Agropecuary Provincial Center of the Diputación of Córdoba in the south of Spain (37°54′50.9″ N–4°42′40.4″ W). The animals were kept in the natural photoperiod in an area of 95 × 95 × 95 cm individually. The roosters were provided with water ad libitum and a commercial diet consisting of 20.00% crude protein, 1.10% lysine, 0.55% methionine, 3.50% crude fat, 5.00% crude fiber, 0.80% calcium, 0.60% phosphorus, and 0.30% salt.

### 2.3. Semen Collection

Semen samples were collected from roosters using abdominal massage [36] twice a week between March and April 2022, obtaining a total of six ejaculates per animal. Semen samples were collected individually in Eppendorf tubes and placed in a water bath at 37 °C to finally make a pool for each working day with those ejaculates that met the minimum quality criteria, established as 0.2–0.6 mL volume, ≥3 × 10^9^ spermatozoa/mL sperm concentration, ≥80% motility, and ≤10% abnormal morphology.

### 2.4. Isolation of Pectoliv30

A prototype steam treatment reactor designed by the Food Phytochemistry Department of the Instituto de la Grasa (Seville, Spain) was used for the hydrothermal treatment of alperujo. Fresh alperujo samples were treated with saturated steam, which was injected directly to increase the contact with the alperujo for 30 min at 160 °C, and the pressure was subsequently reduced in a controlled manner to reach atmospheric pressure values. In order to separate the solid and liquid phases of the alperujo, the samples were centrifuged at 4700× *g* (Comteifa, S.L., Barcelona, Spain). The liquid phase was then ultrafiltrated at 3000 Da, and the resulting solutions were washed with 5000 mL of water at 40 °C and concentrated under vacuum at 40 °C to obtain a final volume of 150 mL. The fractions recovered with a size greater than 3 KDa were rich in pectic material, which were subjected to precipitation with 85% ethanol and further purified by complexation with EDTA, followed by percolation through a strong resin ion exchanger [37], thus resulting in Pectoliv30 [6].

### 2.5. Extender Preparation

The cryopreservation extender used in this study was composed of 0.20 g of D-(+)-glucose, 3.80 g of D-(+)-trehalose dihydrate, 1.20 g of L-glutamic acid, monosodium salt, 0.30 g of potassium acetate, 0.08 g of magnesium acetate tetrahydrate, 0.05 g of potassium citrate monohydrate, 0.04 g of BES, 0.04 g of Bis-Tris, and 0.001 g of gentamicin sulfate (Fraction A, FA). N-methylacetamide was added to the basic extender at 18% (final concentration 9%) (Fraction B, FB), and a pH of 6.8 and osmolarity of 360 mOsm/kg were set [38]. To determine the effect of Pectoliv30, the diluent in question was supplemented with different concentrations of the antioxidant, obtaining four treatments according to this concentration: T1 (extender without antioxidant), T2 (100 µg/mL), T3 (200 µg/mL), and T4 (400 µg/mL). To obtain the desired concentrations, the concentration of the antioxidant required for each treatment was divided into two parts that corresponded to each fraction of the diluent described earlier.

### 2.6. Sperm Freezing and Thawing

After collection, the semen samples were refrigerated for 1 h until reaching 5 °C in a programmable cooler (Cell Incubator SH-020S; Welson, Republic of Korea), aliquoted, and diluted with FA of the diluent according to the experimental design to obtain different treatments. Thirty minutes after the first dilution, a second dilution was performed with FB, and ten minutes later, the semen was packed into 0.25 mL straws to obtain a final concentration of 250 × 10^6^ spz/straw. After 30 min from the second dilution, the straws were placed over nitrogen vapor at a height of 4.5 cm above liquid nitrogen for 30 min. Finally, the straws were immersed in liquid nitrogen for storage. For thawing, the straws were placed in a water bath at 5 °C for 100 s [38]. One sample per treatment was analyzed for each replicate.

### 2.7. Assessment of Sperm Quality

#### 2.7.1. Sperm Motility

For the measurement of sperm motility and kinematic parameters, a CASA IVOS 12.3 (Hamilton-Thorne Bioscience, Beverly, MA, USA) was used [39]. For this purpose, the sperm sample was diluted with FA to a concentration of 50 × 10^6^ spz/mL, and 10 µL was placed in a fixed-height Life Optic Chamber at 37 °C. To be considered spermatozoa, the cells analyzed must reach an area of 2 to 60 µm. Spermatozoa were determined to have progressive movement when VAP > 50 μm/s and STR > 70%. The parameters analyzed were as follows: total motility (TM, %), progressive motility (PM, %), curvilinear velocity (VCL, μm/s), straight-line velocity (VSL, μm/s), average path velocity (VAP, μm/s), straightness (STR, %), linearity (LIN, %), amplitude of lateral head displacement (ALH, μm), and beat/cross frequency (BCF, Hz). The measurement of these parameters was performed objectively using the software integrated into the CASA system. Ten fields were analyzed for each sample to obtain an average for each parameter evaluated.

#### 2.7.2. Sperm Morphology

Eosin-nigrosin staining was used to study sperm morphology [40]. For the preparation of this stain, 0.67 g of eosin Y (Panreac, Barcelona, Spain) and 0.90 g of sodium chloride (Panreac, Barcelona, Spain) were dissolved in 100 mL of bi-distilled water, and then, 10 g of nigrosin was added (Panreac, Barcelona, Spain) to this mixture. For the evaluation of morphology, the sperm sample was diluted with FA to obtain a concentration of 50 × 10^6^ spz/mL. On a slide, 10 μL was deposited and mixed with 10 µL of the stain, followed by a smear test. Finally, when the stain was dry, a total of 200 spermatozoa were counted under the microscope at 1000× magnification (Olympus, Tokyo, Japan), and the percentage of morphologically normal spermatozoa was determined.

#### 2.7.3. Membrane Functionality (HOST)

Sperm plasma membrane functionality was determined with the HOST [41]. For the preparation of this solution (100 mOsm/kg), 1 g of sodium citrate was dissolved in 100 mL of bi-distilled water. Next, 25 µL of semen was mixed with 500 µL of HOST solution, and incubation was carried out at 37 °C for 30 min. After this time, 500 µL of 2% glutaraldehyde was added to fix the spermatozoa, a total of 200 sperm were counted using phase contrast microscopy (×400 magnification), and we determined what percentage had an intact and functional membrane.

#### 2.7.4. Flow Cytometry Analysis

Sperm viability, acrosome integrity, ROS, LPO, and glutathione parameters were analyzed by flow cytometry using the CyFlow^®^ Cube 6 Cytometer (Sysmex Europe GmbH, Hamburg, Germany). The cytometer consists of a 488 nm blue laser and a 638 nm red laser and is equipped with interchangeable optical filters. The cell population was previously defined by granulometric selectivity gates (FSC) and spore granularity (SSC) without using a fluorochrome, and a maximum of 10,000 events were analyzed in each evaluation. For this purpose, the device presents three fluorescence channels (FL1, 536/40 nm bandpass; FL2, 570/50 nm bandpass; FL3, 675 nm low pass), together with forward scatter detection (FSC, trigger parameter) and side scatter detection (SSC). All the protocols used had been previously adapted to the avian species, due to the peculiarities of rooster spermatozoa and the lack of a specific protocol for the avian species [42,43,44].

##### Viability

This parameter was measured using the LIVE/DEAD^®^ sperm viability kit (Molecular Probes Europe, Leiden, The Netherlands). For this purpose, 200 µL of semen (20 × 10^6^ spz/mL) was deposited in a cytometer tube, to which 300 µL of cytometer buffer was added. Next, 5 µL of SYBR-14 (2 µM) and 20 μL of propidium iodide (PI, 480 µM) were added, and the mixture was incubated for 15 min in darkness. After this time, 1200 µL of cytometer buffer was added for measurement. Spermatozoa with intact plasma membranes emitted at the green wavelength (FL1).

##### Acrosome Integrity

For the analysis of this parameter, a 300 µL (20 × 10^6^ spz/mL) semen sample was deposited in a cytometer tube. Subsequently, 30 μL of PI (6 µM) (Molecular Probes Europe, Leiden, The Netherlands) and 15 µL of peanut agglutinin conjugated with fluorescein isothiocyanate (PNA-FITC, 100 µg/mL) (Sigma-Aldrich, St. Louis, MO, USA) were added. After incubating the sample for 5 min in the dark, 1200 µL of cytometer buffer was added for evaluation. Spermatozoa with intact plasma membranes and acrosomes were those that were not stained by these fluorochromes.

Due to the assumption that certain samples selected as a population do not correspond to spermatozoa, the results obtained for viability and acrosome integrity parameters were corrected according to those proposed by Petrunkina et al. [45].

##### Reactive Oxygen Species (ROS)

The study of this parameter was carried out with the commercial DCFH-DA kit (Sigma-Aldrich, St. Louis, MO, USA). For this purpose, 1 µL of DCFH-DA (25 µM) was added to a cytometer tube already containing 1000 µL of semen (4 × 10^6^ spz/mL). After incubation in darkness for 30 min, the sample was centrifuged at 2600 rpm for 5 min before measurement. After removing the supernatant, 1000 µL of cytometer buffer was added for the measurement of ROS.

##### Lipid Peroxidation (LPO)

C11-BODIPY^581/591^ (Molecular Probes Europe, Leiden, The Netherlands) was used for the study of lipid peroxidation. To this end, 10 µL of this fluorochrome was added to a cytometer tube containing 200 µL of semen (20 × 10^6^ spz/mL). The sample was incubated in darkness for 30 min at 37 °C and centrifuged at 2600 rpm for 5 min. Finally, before measurement, the supernatant was removed, and 1000 µL of cytometer buffer was added.

##### Glutathione

For the evaluation of this parameter, 1000 µL of semen (4 × 10^6^ spz/mL) was deposited in a cytometer tube. Next, 0.50 µL of CMFDA CellTracker™ (Molecular Probes Europe, Leiden, The Netherlands) (5 uM) was added, and the mixture was incubated for 30 min at 37 °C in the dark. The sample was then centrifuged at 2600 rpm for 5 min, the supernatant was removed, and 1000 µL of cytometer buffer was added for evaluation.

### 2.8. Statistical Analysis

#### 2.8.1. Overall Descriptive Statistics

The means of each cryopreservation treatment group and fresh samples were established for all sperm-quality-related traits studied in this work: TM, PM, VAP, VSL, VCL, ALH, BCF, STR, LIN, morphology, HOST, glutathione, ROS, LPO, viability, and acrosome integrity. To perform this analysis, the descriptive routine of the data description feature of XLSTAT software (Addinsoft Pearson Edition 2021, Addinsoft, Paris, France) was used.

#### 2.8.2. Data Mining CHAID DT

Following the methodology proposed by González Ariza et al. [46], a data mining CHAID DT was developed as a tool that allows data classification, prediction, interpretation, and manipulation. For decision making, there is an algorithm that includes a node, root, branches, and leaf nodes. To construct each internal node, we started from an observational trait, which in our case corresponded to the semen quality parameters analyzed, provided that a chi-square test significance split criterion (*p* < 0.05) was met (pre-pruning). According to Breiman et al. [47], in order to ensure that the trees do not present a large number of branches and that they do not pursue branches that may contribute significantly to the overall fit, pruning processes should be applied either before or after. Once the tree has been calculated and exhaustively represents the significant relationship between the independent variables detected, the next step is to discard those nodes that do not contribute to the overall prediction. Given the complexity of the model, this statistical method adds an element of penalty, so Bonferroni inequality is used to significantly adjust significance levels. Breiman’s method is similar to forward stepwise regression, but instead of using F-at-entry-based tests, it uses chi-squared tests, which allows the final number of steps to be trimmed. In this sense, each branch represents two or more test results, and each leaf node or terminal node represents a category level of the target variable that corresponds to the semen quality parameters. The top node of the tree is the root node. Decision making is performed through each individual node, and each data record continues until a terminal node of the tree is reached [48]. The tree routine of the analyzing data feature of XLSTAT software, version 2022.4.1 (Addinsoft Pearson Edition 2021, Addinsoft, Paris, France) was used.

For the construction of the CHAID DT in this study, the dependent variable used was the four treatments used for freeze-thawing semen (T1, T2, T3, and T4). The independent variables introduced for the analysis were all semen quality parameters: TM, PM, VAP, VSL, VCL, ALH, BCF, STR, LIN, morphology, HOST, glutathione, ROS, LPO, viability, and acrosome integrity.

#### 2.8.3. Correlation Matrix

A correlation matrix between the different semen quality parameters analyzed was created and graphically represented through a color map elaborated through the web server Heatmapper (www.heatmapper.ca; accessed on 30 March 2024).

#### 2.8.4. CHAID DT Cross-Validation

Ten-fold cross-validation was conducted to ensure that the set of predictors considered significantly explained the differences across the dependent variable groups and to validate the outcomes of the CHAID decision tree. All sample records from the training sample and the study data were used to perform the ten-fold cross-validation. Ultimately, the aim was to determine whether the shortest tree efficiently and repeatably captures the largest number of significant relationships. To perform the cross-validation, we compared the differences between the prediction error of a tree applied to a new sample (resubstitution/replacement error rate) and a training sample (cross-validation error rate). On the one hand, the cross-validation error rate (risk) was the average of the risks of the 10 test samples (folds, new samples), which determined the accuracy of the model. This process was repeated for each fold, and an estimate of the inter-fold error was calculated. Finally, the tree that produced the lowest cross-validation error rate and, therefore, presented the best fit was selected. On the other hand, the substitution error rate was defined as the proportion of misclassified original observations by several subsets of the original tree, which decreased as the tree depth increased. Ultimately, the optimal tree depth was determined at the shallowest tree whose cross-validation risk did not exceed the risk of the minimum-cross-validation-risk tree plus one standard error. To ensure this, the resubstitution error rate and the cross-validation error rate must be similar.

## 3. Results

### 3.1. Overall Descriptive Statistics

The mean for each cryopreservation treatment and for fresh samples in each seminal-quality-related group is shown in Table 2. In general, treatments in which Pectoliv30 was used (T2, T3, and T4) reported more desirable results than the control treatment (T1).

### 3.2. Correlation Matrix

The results of the correlation matrix between the semen quality traits are shown in Figure 1 and had values between +0.949 and −0.467. This heat map represents the correlation matrix between the quality-related traits that were evaluated, taking into account all the cryopreservation treatments (T1, T2, T3, and T4).

The correlation matrix shows the correlation coefficients of different variables. The matrix shows how all possible pairs of values in a table are related to each other. It is a powerful tool for summarizing a large data set and finding and displaying patterns in it. The correlation matrix can be used to determine which variables are significantly connected to each other and which are poorly or not at all correlated. This information can be used to create informed forecasts and judgments based on the facts and therefore would allow for optimization of time and work dynamics.

A high positive correlation was described between VSL and VAP (+0.949), LIN and STR (+0.911), VSL and PM (+0.864), LPO and ROS (+0.808), VAP and PM (+0.806), TM and PM (+0.795), VAP and VCL (+0.780), ROS and glutathione (+0.739), and LPO and glutathione (+0.704). In contrast, there was a negative correlation between yjr BCF and TM (−0.467), viability and ROS (−0.432), BCF and PM (−0.412), acrosome integrity and LPO (−0.411), viability and LPO (−0.381), VAP and BCF (−0.375), and STR and VCL (−0.375).

### 3.3. Data Mining CHAID DT

The data mining CHAID DT obtained from the chi-square dissimilarity matrix is represented in Figure 2. The distribution between the branches and nodes suggested significant differences between treatments for the HOST variable, classifying the observations into three groups (≤64.25, 64.25–75, >75). The HOST variable defined the terminal node and gave rise to leaf nodes. Considering the 24 samples used in the study, only 5 of them had HOST values above 75. Of these five samples with a value above 5, four of them corresponded to T4. Therefore, T4 obtained four samples with HOST values greater than 75 and two with values between 64.25 and 75.

## 4. Discussion

In the poultry industry, the reproductive capacity of roosters must be determined. It is necessary to control various seminal quality traits, which, in turn, will determine fertility, allowing for the implementation of effective artificial insemination programs [49].

By means of the correlation matrix, we can determine the correlations between the quality variables contemplated in our study, which provides us with useful information.

Overall, there is a correlation between the variables VSL, VAP, and PM. The quality of sperm motility is determined by the different kinematic parameters that allow differentiating spermatozoa subpopulations according to the type of movement and speed they have [50]. In the case of VSL, this trait is estimated to be the most accurate parameter for determining sperm velocity [51], and VAP is related to a high fertility rate in goats [52], being also an important predictor of the fertilizing potential of bull semen for both fresh and frozen sperm [53]. The high correlation between both parameters has been previously observed in a study in Japanese quail [54]. The fact that PM and the various velocity parameters were found to be positively correlated may indicate that those spermatozoa presenting a straight, forward linear trajectory will be able to follow a given trajectory in a shorter time.

Furthermore, VAP and VCL were positively correlated in another study in swine in which the aim was to predict the litter size by analyzing motility and different kinematic parameters [55]. Both VCL and LIN are parameters that are related to hyperactive sperm motility in humans, rats, and pigs. An optimal movement pattern is fundamental for fertilization in mammals as it causes vigorous flagellar movement [56,57,58].

The high correlation between LIN and STR has also been observed in hamster sperm in which elevated STR values at the onset of sperm hyperactivation were followed by a gradual rise in LIN values [59]. Both parameters behave similarly [60,61,62] and therefore, have a positive correlation. The correlation between TM and PM was also strong since these two traits have always been linked to each other as they tend to behave in the same way [63]. Lastly, the vast majority of correlations between the kinematic parameters observed in our study coincide with those reported by Kathiravan et al. [64] in bulls, where these parameters were evaluated to predict fertility.

In short, the addition of antioxidants to the cryopreservation diluent impacts motility parameters in rooster sperm. During the cryopreservation process, there is a gradual decrease in energy, due to improved ATP production as the integrity of the mitochondria deteriorates. By adding antioxidants, this effect can be slowed down, which will allow the spermatozoa to move properly, allowing fertilization [65].

The fact that LPO and ROS are correlated is because high ROS production results in high LPO, which leads to decreased sperm motility, DNA damage, and reduced sperm plasma membrane integrity [66,67,68]. In this sense, the addition of antioxidants to the cryopreservation extender is beneficial in reducing the concentration of ROS and thus LPO. This fact has been observed in numerous species, such as cattle [69], goats [70], and horses [71]. Regarding roosters, Amini et al. [72] reported that supplementation of the extender with catalase reduces the level of LPO, with a consequent improvement in sperm motility and viability. Likewise, reduced glutathione supplementation reduces the level of LPO [73]. Quercetin supplementation of a casein-based extender also results in improved semen quality in roosters as it allows upregulation of endogenous enzyme activities, resulting in decreased ROS production [74].

Glutathione plays a fundamental role in maintaining intracellular redox conditions and acts as an antioxidant for both endogenous and exogenous compounds [75]. The fact that glutathione correlates positively with both ROS and LPO is surprising since this substance favors ROS reduction, thanks to its thiol group, which can react against toxic H_2_O_2_ and hydroperoxides [76]. However, it has been shown that too high levels of glutathione impair cellular calcium exchange, leading to alterations in homeostasis, which would damage spermatozoa, being in this case more sensitive to LPO that occurs during the cryopreservation process, which in this case would explain the positive correlation between glutathione with ROS and LPO [75].

The negative correlation between BCF with TM and PM has been previously observed in Venda roosters [77]. BCF also decreases in ostriches with higher mass sperm motility scores [78]. BCF is an important indicator of the fertilizing capacity of spermatozoa, with sperm needing a high beating frequency to be able to ascend through the female reproductive tract and thus complete fertilization [79]. A negative correlation of this parameter with VAP is also observed with both TM and VAP, and this fact can be explained since there is an inverse relationship between the velocity parameters (VSL, VCL, and VAP) and their derivatives (STR, LIN, and BCF) as they are inversely proportional to each other [80]. When the sperm travels a trajectory, as BCF increases, the curvilinear motion of the sperm decreases, and there is a negative correlation between VCL and PM and, finally, between PM and TM. This same theory explains the negative correlation between STR and VCL.

Sperm viability decreases as ROS and LPO increase. The plasma membrane of rooster spermatozoa has a high amount of polyunsaturated fatty acids (PUFAs), which are sensitive to lipid peroxidation due to the high amount of ROS produced during the cryopreservation process, negatively influencing sperm viability [81]. In fact, the main cause of altered sperm functionality is lipid peroxidation, especially in terms of loss of fluidity or damage to the plasma membrane, which ultimately affects sperm viability [82]. Both in mice [83] and in human spermatozoa [84], exogenous production of ROS results in a reduction in sperm viability by 13% and 10–20%, respectively. In addition, supplementation of the extender with exogenous antioxidants, such as quercetin, has been shown to improve sperm viability in roosters [85].

The acrosome is a critical structure for fertilization as a proper acrosomal reaction must occur, which is also disadvantaged during the cryopreservation process [86]. During the cryopreservation process, ROS result in acrosomal alterations, inducing premature capacitation, acrosomal reaction, or damage to acrosome integrity [87]. The use of antioxidants in cryopreservation extenders, such as reduced glutathione, has been shown to improve acrosome integrity in roosters, which could be due to slowing down of the existing lipid peroxidation [73].

In short, antioxidants, such as Pectoliv30, can improve the quality of thawed sperm, which can be evaluated using various parameters considered in this study. The data mining CHAID DT obtained from the chi-square dissimilarity matrix suggests that there are only differences between treatments for the HOST variable. Using this test, we can predict the damage that the plasma membrane of the spermatozoa has suffered during the cryopreservation process, due to the loss of permeability that occurs. The plasma membrane is a key structure for the survival of spermatozoa in the female reproductive tract by acting as a selective barrier between the intracellular and the extracellular medium, so the condition of the plasma membrane can predict the fertilizing capacity of spermatozoa [88,89].

The differences between Pectoliv-80A and Pectoliv30 lie in their chemical composition and the way they are obtained. As far as its chemical composition is concerned, Pectoliv30 has a higher amount of uronic acid (42.32 ± 2.52 vs. 29.93 ± 0.92), neutral sugar (46.01 ± 0.45 vs. 30.22 ± 0.92), and mannose (2.48 ± 0.58 vs. 1.82 ± 0.03) and lower amounts, on the contrary, of phenol (5.91 ± 0.77 vs. 10.93 ± 0.32), rhamnose (0.96 ± 0.01 vs. 5.82 ± 0.36), and galactose (0.55 ± 0.03 vs. 13.36 ± 0.07), among others. Regarding the way of obtaining them, Pectoliv30 is obtained after treatment of fresh alperujo samples with saturated steam at 160 °C for 30 min, while for the isolation of Pectoliv-80A, saturated steam is applied at a temperature of 80 °C for 60 min. Both antioxidants affect post-thawing semen quality concerning the HOST, with Pectoliv-80A also showing a significant effect on total motility [24]. In any case, Pectoliv30 leads to better HOST results as its concentration in the cryopreservation diluent increases, reaching 75.85% when T4 (400 µg/mL) is applied. In the case of Pectoliv-80A, better HOST results are also obtained with 72.16% when the highest concentration (500 µg/mL) is applied. However, there is not such a marked difference between the different treatments [24]. In this regard, although Pectoliv-80A has effects on TM, supplementation with Pectoliv30 is of interest as it leads to better HOST results.

Moreover, the discriminatory power of the HOST between groups coincides with that observed in a separate study that evaluated the effect of the addition of HT to the cryopreservation diluent in roosters [23]. In this case, the treatment with a higher concentration of Pectoliv30 (T4) also resulted in higher HOST values. So, high concentrations of this antioxidant provide more desirable results since this treatment allows preserving to a greater extent the functionality of the plasma membrane of the spermatozoa. This coincides with what was observed by Sun et al. [90] when using MitoQ in the cryopreservation diluent in roosters. In this study, high concentrations of the antioxidant improved the HOST variable. This favorable result has also been reported in turkeys with the addition of glutathione to the diluent [91]. However, in other studies in which antioxidants were used in the extender for the cryopreservation of rooster semen, the best HOST results were obtained when medium concentrations of antioxidants were used [92,93,94,95], which suggests that the concentrations in our study could be calibrated to see which is the maximum concentration of antioxidant that produces the best results.

The fact that no differences were observed between treatments for the rest of the variables studied may be due to the reason that the concentrations used may not have had a significant effect. Another possible cause is that irreversible lesions were produced in cellular structures, due to the high oxidative stress generated during the cryopreservation process, and the addition of Pectoliv30 was not effective in preserving the semen quality, since avian spermatozoa are susceptible to this process, due to their low surface-to-volume ratio and the narrowness of their tail [96], to which we must add that the freezability in birds is low if we compare it with that in other mammalian species [97].

## 5. Conclusions

In conclusion, the addition of a high concentration of Pectoliv30 to the cryopreservation extender is beneficial since it produces an improvement in the functionality of the plasma membrane of spermatozoa in roosters. The plasma membrane is a fundamental structure for the survival of spermatozoa in the female reproductive tract, which leads to better fertility results. In addition, the application of Pectoliv30 as an antioxidant is also favorable from the environmental point of view and allows the use of an olive grove by-product, such as alperujo, which is produced in the south of Spain in high quantities and whose accumulation would be dangerous for the environment. However, numerous seminal quality variables are correlated, which would allow the design of a more efficient protocol for the evaluation of seminal quality in roosters. The exclusion of a series of parameters that present a high positive correlation can be discarded, and the process of semen evaluation in roosters could be cheaper and less time-consuming.

## Figures and Tables

**Figure 1 antioxidants-13-01018-f001:**
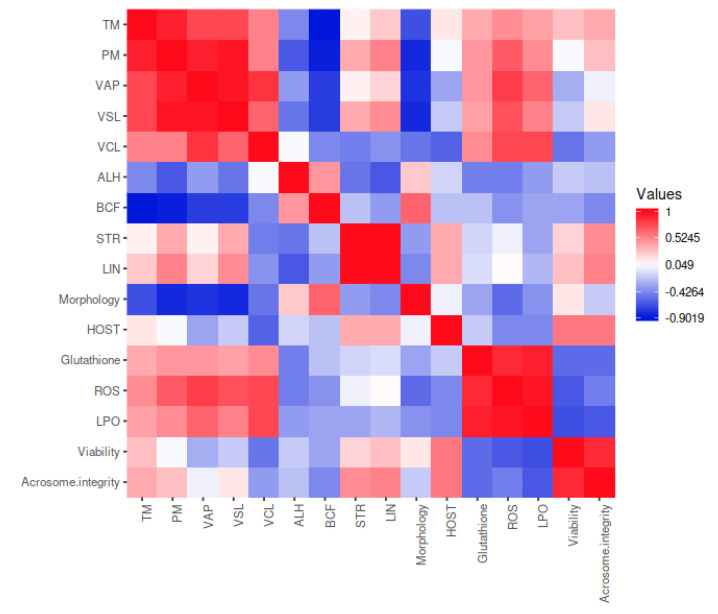
Correlation matrix between the different semen quality traits evaluated in the frozen-thawed semen, taking into account all the cryopreservation treatments in this study (T1: extender without antioxidant; T2: 100 µg/mL of Pectoliv30; T3: 200 µg/mL of Pectoliv30; T4: 400 µg/mL of Pectoliv30).

**Figure 2 antioxidants-13-01018-f002:**
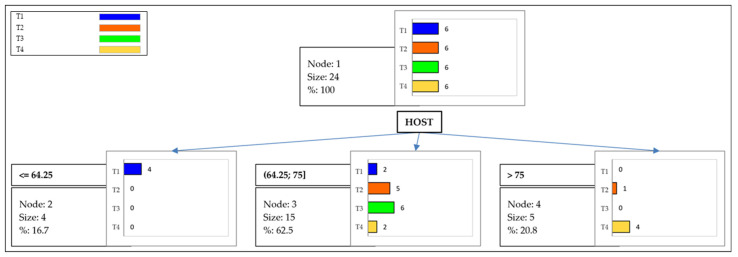
Graphic depiction of the data mining CHAID DT obtained from the chi-square dissimilarity matrix, considering the four freeze-thawing treatments as the clustering criterion (T1: extender without antioxidant; T2: 100 µg/mL of Pectoliv30; T3: 200 µg/mL of Pectoliv30; T4: 400 µg/mL of Pectoliv30). All sperm-quality-related parameters were used in this analysis. Six replicate compounds of all the ejaculates (from 16 roosters used) that met the minimum quality criteria were established.

**Table 1 antioxidants-13-01018-t001:** Chemical composition (g/100 g) and glycosyl residue composition (% molar ratio) of Pectoliv30.

Compound	Quantity (Mean ± SD)
Uronic acid	42.32 ± 2.52
Neutral sugar	46.01 ± 0.45
Phenol	5.91 ± 0.77
Protein	1.24 ± 0.02
Ash	1.42 ± 0.38
Metoxyl + O-acetyl	6.85 ± 0.85
Galacturonic acid	47.91 ± 2.87
Rhamnose	0.96 ± 0.01
Fucose	0.55 ± 0.04
Arabinose	45.37 ± 1.77
Xylose	2.20 ± 0.89
Mannose	2.48 ± 0.58
Galactose	0.55 ± 0.03

**Table 2 antioxidants-13-01018-t002:** Descriptive statistics (mean ± SD) for each cryopreservation treatment group (T1: extender without antioxidant; T2: 100 µg/mL of Pectoliv30; T3: 200 µg/mL of Pectoliv30; T4: 400 µg/mL of Pectoliv30) and fresh samples for all the studied variables in the six replicates used in the study.

	Fresh	T1	T2	T3	T4
TM (%)	69.00 ± 7.13	38.00 ±8.32	43.83 ± 4.79	45.83 ± 5.60	46.67 ± 6.31
PM (%)	23.50 ± 5.21	7.67 ± 2.94	10.00 ± 2.68	11.83 ± 2.79	11.83 ± 3.87
VAP (μm/s)	48.48 ± 3.72	41.42 ± 3.92	42.05 ± 1.68	43.08 ± 2.64	42.92 ± 2.14
VSL (μm/s)	38.48 ± 3.98	32.40 ± 3.02	33.35 ± 1.68	34.75 ± 2.52	34.38 ± 2.84
VCL (μm/s)	78.72 ± 3.94	70.15 ± 6.61	69.62 ± 3.25	68.67 ± 4.68	69.95 ± 1.49
ALH (μm)	3.28 ± 0.26	3.22 ± 0.33	3.10 ± 0.36	2.97 ± 0.37	3.10 ± 0.17
BCF (Hz)	24.90 ± 1.55	25.92 ± 2.40	25.27 ± 2.18	24.15 ± 1.23	23.73 ± 0.74
STR (%)	76.33 ± 2.42	76.17 ± 1.72	77.00 ± 1.67	78.00 ± 2.00	77.50 ± 2.17
LIN (%)	49.83 ± 3.06	50.00 ± 2.10	51.00 ± 2.68	53.17 ± 2.48	51.50 ± 3.15
Morphology (%)	84.94 ± 6.92	72.53 ± 2.56	75.81 ± 3.98	71.10 ± 4.28	70.32 ± 4.32
HOST (%)	81.23 ± 7.62	65.96 ± 4.84	70.49 ± 3.23	71.63 ± 2.74	75.85 ± 4.61
Glutathione (%)	802.33 ± 59.02	991.33 ±182.18	1080.17 ± 131.79	1031.83 ± 151.79	1056.67 ± 113.84
ROS (%)	801.00 ± 28.38	985.50 ± 175.59	984.67 ± 216.18	1034.17 ± 166.59	1013.00 ± 195.16
LPO (%)	858.67 ± 51.33	1026.83 ± 68.66	1028.83 ± 146.08	1011.67 ± 127.87	1049.83 ± 106.04
Viability (%)	90.20 ± 2.84	43.89 ± 7.96	52.61 ± 10.95	49.46 ± 10.55	51.45 ± 6.74
Acrosome integrity (%)	67.58 ± 6.56	18.65 ± 4.64	22.89 ± 6.63	25.26 ± 7.02	28.74 ± 9.63

## Data Availability

All data stemming from the research are contained in the tables and figures. Any additional data can be obtained from the corresponding authors upon reasonable request.

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
