# Peer review of "Effect of Supplementation of a Cryopreservation Extender with Pectoliv30 on Post-Thawing Semen Quality Parameters in Rooster Species"

_antioxidants, 2024, doi:10.3390/antiox13081018_

Round 1

Reviewer 1 Report

I think this manuscript not get the level of Antioxidants. The results is not enough to convince the readers by only showing the heat map. And also, there are lack mechanism studies.

This manuscript is too simple to publish.

Reviewer 2 Report

This study aimed to investigate potential beneficial effects of Pectoliv30, a by-product of olive production and extraction that was been reported to have antioxidant properties, on quality of spermatozoa exposed to freezing-thawing process during semen cryopreservation. Three different concentrations of the Pectoliv30 were added to the cryopreservation extender before freezing of semen samples collected from sixteen Utrerana breed roosters. After thawing, semen quality parameters (sperm motility, sperm morphology, membrane functionality (HOST), sperm viability, and acrosome integrity) and oxidative stress parameters (ROS, lipid peroxidation (LPO), and glutathione) were determined. Statistical analyses included correlation matix between measured parameters and the chi-squared automatic interaction detection (CHAID) decision tree (DT) method. Authors concluded that the highest applied dose of the Pectoliv30 is the most beneficial due to an improvement in the functionality of the plasma membrane of the roosters’ spermatozoa.

This topic could be interesting and important, however there are several key points that need to be made clear.

The main concern I have about this paper is with respect to the statistical methods used in the study and the manner in which the results are presented. The Results section and the Figures are not clear enough to accurately assess the work. The figures should be clear and understandable without reading the text of the manuscript.

1.  Overall descriptive statistics (including the results of the measured parameters obtained before and after freezing-thawing process as well as after the Pectoliv30 treatments) is missing.

2.    Figure 1 is not clear. Is it a correlation matrix of measured parameters in semen before and after freezing/thawing process or Pectoliv30 treatment? Which one of it (since there are four treatments)? How this correlation matrix contributes to answering the research question?

3.    There is a need for more information on the CHAID DT method.  Authors wrote (L 97-100) that “the chi-squared automatic interaction detection (CHAID) decision tree (DT) method is widely used to perform classification, prediction, regression, estimation, description, visualization, and dimensionality reduction, belonging to the non-parametric category”. However, what does it mean? Authors should explain the procedure and guide the readers to their conclusion that “the highest applied dose of the Pectoliv30 is the most beneficial due to an improvement in the functionality of the plasma membrane of the roosters’ spermatozoa”. This is not clear from the results presented. In addition, the name, version and producer of statistical program (or link to webpage) used for this analysis should be added.

4. Results and Figure 2: What is the root node (a dependent categorical variable)? How many independent variables (categorical or numerical) were included in the CHAID procedure? Is only HOST terminal node? 

Reviewer 3 Report

The author has to follow very little corrections so he can improve the manuscript.

Dear author,

Bellow you will find very little corrections that you have to do in your manuscript

Abstract

P1 L21:  For this purpose instead of For this

2. Materials and Methods

2.1. Animal management

P3 L116-117: In this study sixteen Utrerana breeder roosters between one and three years of age were used in this study instead of Sixteen Utrerana breeder roosters between one and three years of age were used in this study.

P4 L182: On a slide 10 μL were deposited instead of 10 μL were deposited on a slide

Discussion

P7 L290-292: In the poultry industry, the reproductive capacity of roosters must be determined. It is necessary to control various seminal quality traits which, in turn, will determine fertility, allowing for the implementation of effective artificial insemination programs instead of In the poultry industry, the reproductive capacity of the roosters has to be determined, being necessary to control the different seminal quality traits which, in turn, will determine fertility, allowing to carry out adequate artificial insemination programs

P8 L319: Lastly, the vast majority of correlations instead of By last, The vast majority of correlations

P8 L322-325: In short, the addition of antioxidants to the cryopreservation diluent impacts motility parameters in rooster sperm. During the cryopreservation process, there is a gradual decrease in energy due to improved ATP production as the integrity of the mitochondria deteriorates instead of In short, the addition of antioxidants in the cryopreservation diluent has an impact on motility parameters in rooster sperm since during the cryopreservation process there is a gradual decrease in energy given the improved production of ATP as the integrity of the mitochondria deteriorates.

P9 L380: various parameters instead of different parameters

Round 2

Reviewer 2 Report

Most of the concerns have been addressed in the revised version of the manuscript and its quality is therefore improved. However, there are still some points that should be improved.

The tables and figures should be clear and understandable without reading the text of the manuscript. It should be clear reading only data presented in table 2 and/or figures 1 and 2 that the presented results are related to the effects of the Pectoliv30 applied in four doses (T1-T4) on fresh and post-thawing semen quality parameters in pooled semen samples obtained from sixteen Utrerana breeder roosters. Applied doses (T1-T4) should be specified. 

Table 1: Results should be presented as mean +/- SD, the number of replicates specified, and measurement units for the presented parameters added.

It is necessary to make sure that expression/names are uniform in whole study. Therefore, in Figure 2, it would be useful to use expression “T1, T2, T3, and T4” instead of “1, 2, 3, 4”.

Figure 2: More details should be provided. It is unclear from this figure what was done in terms of treatments (what different treatments), experimental models used, groups compared, considering parameters, number of animals per group, etc.

In addition, I suggest the authors to define the abbreviations for the measured parameters where it is appropriate.

L 366-368, and L 422-424 belong to results section.

Very similar study of the same group of the authors (Ruiz et al., 2024, https://doi.org/10.1016/j.psj.2024.103630) was recently published. What is the difference between Pectoliv-80A and Pectoliv30? The results of both studies should be compared. Which of these two antioxidants is better at protecting sperm from damage caused by freezing-thawing process?

References 27 and 91 are the same.  
